# Particularities of Older Patients with Obstructive Sleep Apnea and Heart Failure with Mid-Range Ejection Fraction

**DOI:** 10.3390/medicina55080449

**Published:** 2019-08-07

**Authors:** Carmen Loredana Ardelean, Sorin Pescariu, Daniel Florin Lighezan, Roxana Pleava, Sorin Ursoniu, Valentin Nadasan, Stefan Mihaicuta

**Affiliations:** 1University of Medicine and Pharmacy, Dr Victor Babes, Eftimie Murgu Square 2, 300041 Timisoara, Romania; 2Cardiology Department, University of Medicine and Pharmacy, Dr Victor Babes, Eftimie Murgu Square 2, 300041 Timisoara, Romania; 3Department of Public Health and Health Management, University of Medicine and Pharmacy, Dr Victor Babes, Eftimie Murgu Square 2, 300041 Timisoara, Romania; 4Department of Hygiene and Environmental Health, University of Medicine and Pharmacy, Sciences and Technology of Targu Mures, Gheorghe Marinescu 38, 540139 Targu Mures, Romania; 5Pneumology Department, University of Medicine and Pharmacy, Dr Victor Babes, Eftimie Murgu Square 2, 300041 Timisoara, Romania

**Keywords:** obstructive apnea, heart failure, risk factors, elderly, comorbidities

## Abstract

*Background and objectives*: Obstructive sleep apnea syndrome (OSAS) and heart failure (HF) are increasing in prevalence with a greater impact on the health system. The aim of this study was to assess the particularities of patients with OSAS and HF, focusing on the new class of HF with mid-range ejection fraction (HFmrEF, EF = 40%–49%), and comparing it with reduced EF (HFrEF, EF < 40%) and preserved EF (HFpEF, EF ≥ 50%). *Materials and Methods*: A total of 143 patients with OSAS and HF were evaluated in three sleep labs of “Victor Babes” Hospital and Cardiovascular Institute, Timisoara, Western Romania. We collected socio-demographic data, anthropometric sleep-related measurements, symptoms through sleep questionnaires and comorbidity-related data. We performed blood tests, cardio-respiratory polygraphy and echocardiographic measurements. Patients were divided into three groups depending on ejection fraction. *Results*: Patients with HFmrEF were older (*p* = 0.0358), with higher values of the highest systolic blood pressure (mmHg) (*p* = 0.0016), higher serum creatinine (*p* = 0.0013), a lower glomerular filtration rate (*p* = 0.0003), higher glycemic levels (*p* = 0.008) and a larger left atrial diameter (*p* = 0.0002). Regarding comorbidities, data were presented as percentage, HFrEF vs. HFmrEF vs. HFpEF. Higher prevalence of diabetes mellitus (52.9 vs. 72.7 vs. 40.2, *p* = 0.006), chronic kidney disease (17.6 vs. 57.6 vs. 21.5, *p* < 0.001), tricuspid insufficiency (76.5 vs. 84.8 vs.59.1, *p* = 0.018) and aortic insufficiency (35.3 vs.42.4 vs. 20.4, *p* = 0.038) were observed in patients with HFmrEF, whereas chronic obstructive pulmonary disease(COPD) (52.9 vs. 24.2 vs.18.3, *p* = 0.009), coronary artery disease(CAD) (82.4 vs. 6.7 vs. 49.5, *p* = 0.026), myocardial infarction (35.3 vs. 24.2 vs. 5.4, *p* < 0.001) and impaired parietal heart kinetics (70.6 vs. 68.8 vs. 15.2, *p* < 0.001) were more prevalent in patients with HFrEF. *Conclusions*: Patients with OSAS and HF with mid-range EF may represent a new group with increased risk of developing life-long chronic kidney disease, diabetes mellitus, tricuspid and aortic insufficiency. COPD, myocardial infarction, impaired parietal kinetics and CAD are most prevalent comorbidities in HFrEF patients but they are closer in prevalence to HFmrEF than HFpEF.

## 1. Introduction

In recent years, obstructive sleep apnea syndrome (OSAS) has increased in prevalence, occurring in up to 10% of healthy subjects, due to the greater frequency of obesity and the aging of the population. Consequently, this has had an increasingly important impact on the health system [1]. The prevalence of OSAS in subjects with cardiovascular disease, reported in earlier studies, was between 50% and 80% [2,3,4], and in half of subjects with heart failure (HF), it is associated with increased mortality [5] and worse prognosis [6].

OSAS is globally known as a major factor for the occurrence of cardiometabolic comorbidities due to intermittent hypoxia which leads to oxidative stress, endothelial dysfunction, increase of sympathetic activity and systemic inflammation [7]. Furthermore, activation of the sympathetic nervous system leads to activation of the renin-angiotensin-aldosterone system, which increases hydro-saline retention and thus the level of blood pressure [8]. However, hydro-saline retention due to heart failure can also play an important role in the pathogenesis of OSAS [9]. These data suggest that the relationship between HF and OSAS is not fully understood.

Large studies have demonstrated that OSAS prevalence is higher in patients with coronary artery disease (CAD), HF, resistant arterial hypertension associated with risk of stroke, and uncontrolled arrhythmias [10].

Patients with OSAS present a variety of symptoms that correlate with anthropometric measurements, smoking habits, sedentarism and association of comorbidities [1]. In recent years, new perspectives regarding clinical presentations of OSAS with description of different phenotypes and clusters have emerged [11,12,13].

Different structural or functional cardiac abnormalities can lead to occurrence of typical symptoms and signs of HF as defined by the European Society of Cardiology (ESC) guidelines, increased morbidity and mortality and higher costs for the health system [14]. HF is more common in elderly patients, especially those over 60 years [15].

The measurement of the left ventricle ejection fraction (LVEF) is used to define HF. Accordingly, HF is classified as HF with preserved LVEF, ≥50% (HFpEF) and HF with reduced LVEF, <40% (HFrEF). Recently, the latest guidelines on the diagnosis and management of heart failure published by the European Society of Cardiology proposed a new class of HF patients with LVEF = 40%–49% called HF with mid-range EF (HFmrEF), in order to better differentiate HF patients from the point of view of etiology, developing mechanisms and response to treatment strategy [16,17].

## 2. Materials and Methods

### 2.1. Study Subjects

We enrolled consecutive patients evaluated for OSAS at the “Victor Babes” Timisoara Hospital between 2014 and 2018 and for HF at the Timisoara Institute for Cardiovascular Diseases. Inclusion criteria were patients with age over 40 years, with a diagnosis of heart failure and OSAS who performed cardio-respiratory polygraphy, echocardiography and blood test evaluation. Patients with incomplete evaluation and those with no OSAS or with predominantly central sleep apnea (CSA) were excluded. This study was approved by the Ethical Committee of the University of Medicine and Pharmacy “Victor Babes” Timisoara as a subject for a PhD thesis (number 14728/15 NOV 2013). The clinics where the patient’s evaluations were performed have an established agreement with the university through which all the data obtained from the patients may be used for research purposes. Informed consent was signed by all the patients.

Patients were initially evaluated through a standard datasheet with the following parameters: Age (years), gender (male/female), weight (kg) and height (cm), followed by measurement of body mass index (BMI = weight in kg/squared height in m), neck and abdominal circumference (cm), presence and duration of hypertension, maximum and current value of blood pressure, medication, reported apneas, snoring, sleepiness, Epworth Sleepiness Scale, SAS score (sleep apnea syndrome score), morning headache, restless sleep, nocturia, nocturnal awakenings, chronic obstructive pulmonary disease (COPD), diabetes, dyslipidemia, CAD, HF, arrhythmias, stroke, nasal septum deviation, polyposis, hypertrophic uvula and smoking status (pack × years). Since it is not routine practice in our cardio-respiratory unit, we did not collect data about physical activity.

For the sleep study we followed the European standards for diagnosis of OSAS [18].

Cardio-respiratory polygraphy recording was performed with Stardust Respironics and Porti. Several parameters were measured: The number of apnea (individually, central, obstructive and mixed) and hypopnea per hour of sleep and per night, the AHI (apnea-hypopnea index), the desaturation index, the mean saturation, the lowest saturation, and the longest desaturation period below 88% (seconds). Because we did not perform full night assisted polysomnography, data about sleep duration and duration of the lowest desaturation were not recorded. Approximately one-third of the patients enrolled in this study used CPAP (continuous positive airway pressure) due to the non-reimbursement of the cost of this therapy. Therefore, data related to the use of CPAP have not been included in this study.

The cardio-respiratory polygraphy recording was performed and scored manually as stated by American Academy of Sleep Medicine standards and European Sleep Research Society recommendations [19].

Laboratory tests were performed in Romanian Accreditation Association-RENAR certified medical laboratories, as follows: ESR (erythrocyte sedimentation rate) (mm/h), glucose (mg/dL), uric acid (mg/dL), creatinine (mg/dL), erythrocyte count (×10⁶/μL), hemoglobin (g/dL), sodium (mmol/L), potassium (mmol/L) and lipid profile (total cholesterol, LDL (low-density lipoprotein)-cholesterol, HDL (high-density lipoprotein)-cholesterol, triglycerides, mg/dl). Glomerular filtration rate, (GFR, mL/min/1.73 m^2^) was calculated for each patient, using CKD-EPI (Chronic Kidney Disease Epidemiology Collaboration) formula [20]. Blood samples were collected early in the morning after fasting, and within 1–2 days of informed consent if signing took place at a different time of day.

Cardiological evaluation was performed, for all patients, at the Institute of Cardiovascular Diseases in Timisoara, using the same diagnostic algorithm and equipment. We used the modified Simpson’s rule for echocardiographic measurement of EF [21], and HF was classified depending on the LVEF, HF with preserved ejection fraction, LVEF ≥ 50% (HFpEF); HF with reduced ejection fraction, LVEF < 40% (HFrEF); and HF with mid-range ejection fraction, LVEF = 40%–49% (HFmrEF). We also recorded end-diastolic volume (mL), end-systolic volume (mL), left atrium surface (cm^2^), left atrium diameter (cm), right ventricle diameter (cm), mitral E and A wave (m/s), E/A ratio, pulmonary artery pression (mm Hg) and percentage of patients with impaired parietal heart kinetics. Although, the assessment of the left ventricular internal dimension, left ventricular posterior wall, interventricular septum thickness is performed in current practice and provides valuable information about the HF etiology, in this study they were not recorded because we enrolled patients with heart failure, regardless of the underlying cardiac disease. We studied patients regarding LVEF only, as the main cardiac marker.

The morphological aspect, area (cm), degree of regurgitation and stenoses and transvalvular pressure gradients were determined for the mitral, aortic, tricuspid and pulmonary valves [22].

### 2.2. Statistical Analysis

Data are presented as proportions, medians and interquartile range (IQR) for variables with a skewed distribution. The differences in the characteristics of the subjects were evaluated after being divided into three groups, depending on the EF (EF < 40%, EF = 40%–49%, EF ≥ 50%). We used the chi-squared test (two degrees of freedom) for comparison of categorical data between groups of patients. Continuous data were tested for normality using the Kolmogorov–Smirnov test. Data with non-normal distributions were compared using the Kruskal–Wallis test. The *p* values for all hypothesis tests were two-sided, and the *p* value was set to the statistical significance threshold of <0.005. All data analyses were performed with Stata 15.1 (Statacorp, TX, USA).

## 3. Results

A total of 143 patients with OSAS and HF were evaluated in three sleep labs of Timisoara “Victor Babes” Hospital, Western Romania.

### 3.1. Socio-Demographic and Anthropometric Data

Patients were divided into three groups depending on EF, with the following characteristics, presented as median and interquartile range: 17 patients (11.88%) with HFrEF, of which 15 male (88%), age 61 (56–69) years, BMI 35 (31–36) kg/m^2^, neck circumference 44 (39–46) cm, abdominal circumference 120 (114–128) cm; 33 patients (23.07%) with HFmrEF, of which 22 male (67%), age 64.5 (57.5–71) years, BMI 36 (31.5–41.5) kg/m^2^, neck circumference 45 (42–46) cm, abdominal circumference 120 (114–130) cm; 93 patients (65.93%) with HFpEF, of which 62 male (67%), BMI 35 (31–41) kg/m^2^, neck circumference 44 (41–46) cm, abdominal circumference 122 (115–130) cm (Table 1).

Patients from the HFmrEF group were significantly older. More males were found in the HFrEF group. There were no differences in terms of BMI, neck and abdominal circumference (Table 1).

### 3.2. Sleep Study and Blood Pressure Data

There were no differences between groups of patients regarding blood pressure (BP) measurement and sleep study, systolic and diastolic BP at visit, AHI, type of apneas, desaturation index, medium and lowest desaturation, longest desaturation <88% and sleep questionnaire. Significant differences were observed in patients with HFmrEF regarding the highest systolic BP reported by the patients (*p* = 0.016) (Table 2).

### 3.3. Blood Tests

Routine blood tests revealed significant statistical difference in HFmrEF patients regarding level of glucose (*p* = 0.0081), creatinine (*p* = 0.0013) and GFR (*p* = 0.0003) (Table 3). There were no differences for ESR, uric acid, erythrocytes, hemoglobin, Na, K, total cholesterol, LDL and HDL cholesterol, or triglycerides.

### 3.4. Echocardiographic Measurements

Regarding echocardiographic measurements, statistically significant differences were found for end-diastolic and end-systolic volumes, ejection fraction, and left atrial diameter. LA (left atrium) diameter was higher in patients with HFmrEF (*p* = 0.0002), similar to other publications (Table 4)

### 3.5. Comorbidities

Regarding comorbidities, data were presented as proportions, HFrEF vs. HFmrEF vs. HFpEF. We observed that the group with HFmrEF has significantly more cases of diabetes mellitus (52.9 vs. 72.7 vs. 40.2 *p* = 0.006), chronic kidney disease (17.6 vs. 57.6 vs. 21.5, *p* < 0.001), valvular disease, tricuspid insufficiency (76.5 vs. 84.8 vs. 59.1, *p* = 0.018) and aortic insufficiency (35.3 vs. 42.4 vs. 20.4, *p* = 0.038). The group with HFrEF had more cases of COPD (52.9 vs. 24.2 vs. 18.3, *p* = 0.009), myocardial infarction (35.3 vs. 24.2. vs 5.4, *p* < 0.001), CAD (82.4 vs. 66.7 vs. 49.5, *p* = 0.026) and impaired heart parietal kinetics (70.6 vs. 68.8 vs. 15.2, *p* < 0.001). The presence of myocardial infarction, CAD and impaired heart parietal kinetics were much lower in HFpEF patients compared with HFmrEF and HFrEF (Table 5).

## 4. Discussion

In our population, 23.07 % of the patients had HFmrEF, higher than reports from recent studies where the percentage of the HFmrEF category is between 13% and 17% [23,24,25,26].

Men are more likely to have OSAS in patients with HF. Moreover, men have a higher incidence of HF in patients with OSAS [27]. In our study, patients with HFmrEF were older, with no significant differences regarding gender or neck and abdominal circumferences.

It is well known that obesity is an important risk factor for heart failure, and this association leads to multiple complications. In addition, obesity seems to be more prevalent in HF patients with preserved ejection fraction; this may occur due to poor echocardiographic images and error in LVEF measurement [28]. In our study, we included only patients with OSAS, and patients with HFmrEF were in stage 2 of obesity, with higher BMIs, but differences were not statistically significant. Central sleep apnea (CSA) is particularly noted in patients with HFrEF, and decompensated HF has been recognized as a risk factor for CSA [29].

Some studies have demonstrated that patients with heart failure and OSAS are less symptomatic, regardless of AHI, and Epworth Sleepiness Scale does not correlate with AHI [30]. Questionnaires do not accurately predict OSAS in patients with cardio-vascular disease (CVD) [31]. Epworth Sleepiness Scale and SAS score can be beneficial in predicting OSAS, but in our groups of patients, although the values are high, differences between groups are insignificant [32].

In our group, all the patients have severe OSAS, regardless of EF. Patients with severe, untreated OSAS have a higher risk of fatal cardiovascular events, some studies show [33].

Our patients with HFmrEF have higher blood glucose, serum creatinine and decreased glomerular filtration rate.

Nielson demonstrated in a large study that patients with elevated blood glucose levels but without confirmed diabetes have an increased risk of developing HF. Therefore, these patients should be carefully monitored in order to prevent the onset of HF [34].

Many studies demonstrated that even mild impaired renal function, with transitory elevated level of serum creatinine, represents an important predictor for worsening of heart failure. The pathophysiology remains unclear, but venous congestion and intrabdominal pressure serve as a challenge for the development of new therapeutic approaches [35,36]. OSAS severity was correlated with elevated serum creatinine [37], while CKD stage 3 is considered a significant predictor of CSA, as was demonstrated by Fleischmann et al. [38].

In this study, lipid profile is not different as in a cohort with all severities of disease where OSAS severity was independently correlated with cholesterol and triglycerides levels, probably because all our patients have severe OSAS [39].

Often, patients with HFpEF present only increased wall thickness of the LV or the size of LA, which makes it even more difficult to diagnose. In our study, LA diameter was higher in patients with HFmrEF (*p* = 0.0002), similar to other publications [40]. Moreover, the role of the left atrium in modulating LV function is well-known [41], and there are considerable amounts of data demonstrating that the size of the LA is directly proportional to the increased risk of cardiovascular events; this parameter is not used enough in clinical practice to determine the HF progression [42].

Wang et al. demonstrated in a recent meta-analysis that patients with moderate to severe tricuspid regurgitation (TR) have a higher risk of hospitalization for worsening HF and cardiac mortality. Patients with TR, regardless of severity, have a higher risk of all-cause mortality, compared with patients without tricuspid valvular disease [43]. Asymptomatic patients with HFpEF, but with severe aortic regurgitation (AR), have a higher risk of fatal cardiac events [44].

Comorbidities are very important in HF. Thus, comorbidity management plays a leading role in the treatment and progression of heart failure.

COPD is significantly more prevalent in HFrEF in our population. COPD and OSAS have common pathophysiological mechanisms, such as activation of sympathetic nervous system and inflammation, which can lead to increased cardiovascular risk. Furthermore, patients with association of these diseases, so called “overlap syndrome”, are exposed to an even greater risk [7].

Some patients with advanced stages of COPD have right HF with peripheral edema and have increased likelihood of OSAS because of the shift of the rostral fluid from the legs during the night [45].

Chronic kidney disease is significantly more prevalent in the group of HFmrEF. Reports from ESADA (Sleep apnea network/European sleep apnea database) cohort study identify that in OSAS patients, decrease of GFR was predicted by baseline characteristics like older age, female gender, obese patients and severe nocturnal hypoxemia and by comorbidities like heart failure and arterial hypertension [46].

Several studies reported that HFmrEF patients have an increased risk of CAD as HFrEF patients, but all-cause mortality was similar to HFpEF [47,48]. The prognosis of HF, regardless of EF, was correlated with common risk factors, such as age, underlying disease and comorbidities [49].

Chioncel et al. found that the long-term mortality rate in HFmrEF was between those patients with HFpEF and HFrEF [50], whereas Pascual-Figa et al. showed that HFmrEF patients match a clinical profile similar to HFrEF, with an increased risk of cardiovascular mortality, rather than HFpEF [51]. Still, there are contradictory data from other recent studies which showed that HFmrEF patients have a prognosis similar to HFpEF patients [52,53]. The results of treatment in the latest publication show increased controversies [54].

## 5. Study Limitations

This study has several limitations. The studied population is relatively small, and even smaller for the subjects with HFrEF. There are no data about sleep since we did not perform full-night assisted polysomnography. The results need to be confirmed by larger studies.

## 6. Conclusions

Patients with OSAS and HF with mid-range EF may represent a new group of patients with increased risk of developing life-long chronic kidney disease, diabetes mellitus, and tricuspid and aortic insufficiency. COPD, myocardial infarction, impaired heart parietal kinetics and CAD are the most prevalent comorbidities in HFrEF patients, but the prevalence of these is closer to that of HFmrEF than HFpEF. More studies are needed, on larger groups of patients, to determine how OSAS is involved in the progression of HF, from borderline ejection fraction to more severe heart failure.

## Figures and Tables

**Table 1 medicina-55-00449-t001:** Socio-demographic and anthropometric data.

General Data	EF < 40% (HFrEF) *n* = 17	EF = 40%–49% (HFmrEF) *n* = 33	EF ≥ 50% (HFpEF) *n* = 93	*p*-Value
**Age (years)**	61 (56–69)	64.5 (57.5–71)	61 (56–67)	0.0358
**Male (n0., %)**	15 (88%)	22 (67%)	62 (67%)	0.187
**BMI (kg/m^2^)**	35 (31–36)	36 (31.5–41.5)	35 (31–41)	0.415
**Neck circumference (cm)**	44 (39–46)	45 (42–46)	44 (41–46)	0.6573
**Abdominal circumference (cm)**	120 (114–128)	120 (114–130)	122 (115–130)	0.8569

Data are presented as proportions, medians and interquartile range (IQR). EF, ejection fraction; HFrEF, heart failure with reduced ejection fraction; HFmrEF, heart failure with mid-range EF; HFpEF, heart failure with preserved EF; BMI, body mass index.

**Table 2 medicina-55-00449-t002:** Blood pressure and sleep study.

Sleep/Blood Pressure Parameters	EF < 40% (HFrEF) *n* = 17	EF = 40%–49% (HFmrEF) *n* = 33	EF ≥ 50% (HFpEF) *n* = 93	*p*-Value
**Highest Systolic BP (mmHg)**	161 (161–179)	202 (177.5–220)	191 (170–210)	**0.0016**
**Highest Diastolic BP (mmHg)**	90 (80–100)	100 (90–110)	100 (90–110)	0.1472
**Duration of hypertension (years)**	12 (7–20)	10 (8–14.5)	10 (5–15)	0.3899
**AHI (events/h) **	42 (24–53)	38 (24–48.5)	44 (27–62)	0.1633
**Central apneas**	1 (0.4–9.5)	1.25 (0.25–4.66	1.7 (0.3–5.4)	0.8947
**Obstructive apneas**	13.2 (8.9–21.3)	14.55 (7.2–23.8)	19.2 (12–38.2)	0.0704
**Mixed apneas**	2 (0.9–6)	1.75 (0.45–3.2)	2.1 (0.7–6.1)	0.3153
**Desaturation index**	24 (14.5–51)	30.5 (13.4–46.4)	39.5 (19–53)	0.1856
**Medium SpO2 (%) **	93 (90–94)	92.5 (91–94)	92 (89–93)	0.1403
**Lowest SpO2 (%)**	78 (76–83)	76 (66–83)	77 (62–83)	0.6183
**Longest duration SpO2 <88% (sec)**	50 (21–115)	61 (27–110.5)	83 (30–139)	0.3311
**Epworth Sleepiness Scale**	13 (12–17)	12 (9–15)	13 (10–18)	0.0819
**SAS score **	4.9 (4.5–5.4)	4.3 (4.1–4.85)	4.8 (4.1–5.3)	0.0857

Data are presented as medians and interquartile range (IQR). BP, blood pressure; AHI, apnea-hypopnea index; SpO2, oxygen saturation; SAS score, sleep apneas syndrome score.

**Table 3 medicina-55-00449-t003:** Blood tests.

Blood Tests	EF < 40% (HFrEF) *n* = 17	EF = 40%–49% (HFmrEF) *n* = 33	EF ≥ 50% (HFpEF) *n* = 93	*p*-Value
**ESR (mm/h)**	10.5 (8–25)	15 (8–32)	12 (6–25)	0.4202
**Glucose (mg/dL)**	122.5 (104–130.5)	126 (107–180.5)	108.5 (94–127)	**0.0081**
**Uric acid (mg/dL)**	7 (4.9–8.3)	6.6 (5.2–8.1)	5.9 (5–6.9)	0.2547
**Creatinine (mg/dL)**	1.15 (0.98–1.3)	1.33 (1.13–1.6)	1.075 (0.9–1.33)	**0.0013**
**GFR (mL/min/1.73 m^2^)**	61.8 (58.9–78)	48.8 (38.7–61)	65.7 (51.3–82.3)	**0.0003**
**Erythrocytes(*10⁶/μL)**	4.73 (4.42–5.06)	4.80 (4.45–5.13)	4.77 (4.43–5.12)	0.8399
**Hemoglobin (g/dL)**	14.1 (12–15.9)	14 (12.3–15.1)	14.5 (13.2–15.45)	0.3972
**Na+ (mmol/L)**	140 (138.5–142)	141 (139–144)	141 (139–142)	0.5287
**K+ (mmol/L)**	4.13 (4–4.65)	4.35 (3.9–4.)	4.2 (4–4.6)	0.8157
**Cholesterol (mg/dL)**	166 (143–182.5)	163 (130–195)	164 (135–205)	0.9128
**LDL cholesterol (mg/dL)**	123 (87–132)	94.5 (84–117)	98 (69–132)	0.9003
**HDL cholesterol (mg/dL)**	45 (36–48)	45 (32–54)	43 (37–52)	0.7776
**Triglycerides (mg/dL)**	128.5 (90–166)	96 (82–151)	125 (92–193)	0.3465

Data are presented as medians and interquartile range (IQR). ESR, erythrocyte sedimentation rate; GFR, glomerular Filtration Rate; LDL, low-density lipoprotein; HDL, high-density lipoprotein.

**Table 4 medicina-55-00449-t004:** Echocardiographic measurements.

Echocardiography Parameters	EF < 40% (HFrEF) *n* = 17	EF = 40%–49% (HFmrEF) *n* = 33	EF ≥ 50% (HFpEF) *n* = 93	*p*-Value
**End–diastolic volume (ml)**	185 (140–220)	118 (94–155)	130 (110–147.5)	**0.0027**
**End–systolic volume (ml)**	123.5 (90–154)	64.9 (53–84.5)	60 (48.5–65.5)	**0.0001**
**EF (%)**	31.58 (30–35.71)	44.87 (43.37–46.13)	55 (50.98–59)	**0.0001**
**LA surface (cm^2^)**	27 (21–32)	28 (24.5–31)	25 (23–29)	0.4666
**LA diameter (cm)**	4.7 (4.6–5)	4.95 (4.5–5.3)	4.3 (3.9–4.64)	**0.0002**
**RV diameter (cm)**	3.24 (2.5–3.6)	2.9 (2.6–3.25)	2.8 (2.5–3.14)	0.2684
**Mitral E wave (m/s)**	0.74 (0.63–1)	0.76 (0.58–1.07)	0.73 (0.55–0.9)	0.298
**Mitral A wave (m/s)**	0.70 (0.5–1.07)	0.8 (0.6–1)	0.7 (0.6–0.9)	0.6517
**E/A ratio**	0.86 (0.74–1.65)	0.79 (0.64–1.4)	0.87 (0.73–1.3)	0.6883
**PAP (mmHg)**	47.5 (25–63.3)	45 (34.5–50)	36 (25.5–45.5)	0.1303

Data are presented as medians and interquartile range (IQR). LA, Left atrium; RV, right ventricle; PAP, pulmonary artery pressure.

**Table 5 medicina-55-00449-t005:** Comorbidities.

Comorbidities %	EF < 40% (HFrEF) *n* = 17	EF = 40%–49% (HFmrEF) *n* = 33	EF ≥ 50% (HFpEF) *n* = 93	*p*-Value
**Hypertension**	100	97	95.7	0.670
**Smoking**	35.3	15.2	36.6	0.070
**COPD**	52.9	24.2	18.3	**0.009**
**Diabetes mellitus**	52.9	72.7	40.2	**0.006**
**Dyslipidemia**	76.5	66.7	66.7	0.719
**Atrial fibrillation**	58.8	57.6	37.6	0.065
**Stroke**	5.9	6.1	12.9	0.781
**Myocardial infarction**	35.3	24.2	5.4	**<0.001**
**CAD**	82.4	66.7	49.5	**0.026**
**CKD**	17.6	57.6	21.5	**<0.001**
**Mitral insufficiency**	94.1	90.9	77.4	0.088
**Tricuspid insufficiency**	76.5	84.8	59.1	**0.018**
**Aortic insufficiency**	35.3	42.4	20.4	**0.038**
**Pulmonary insufficiency**	17.65	15.63	12.9	0.840
**PAH**	47.1	39.4	25.8	0.298
**Pulmonary thromboembolism**	5.9	3.0	5.4	0.863
**Impaired parietal heart kinetics**	70.6	68.8	15.2	**<0.001**

COPD, chronic obstructive pulmonary disease; CAD, coronary artery disease; CDK, chronic kidney disease; PAH, pulmonary arterial hypertension.

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
