# Peer review of "Particularities of Older Patients with Obstructive Sleep Apnea and Heart Failure with Mid-Range Ejection Fraction"

_medicina, 2019, doi:10.3390/medicina55080449_

Round 1

Reviewer 1 Report

Comments

1)    Why didn’t collected data for physical activity?

2)    Why didn’t assessment the left ventricular internal dimension, left ventricular posterior wall, interventricular septum thickness.

3)    It would be beneficial to have data about duration of the minimum SpO2 during PSG study.

4)    Which percent of participants had diabetes mellitus?

5)    Please add the sleep duration and the percent of patients which used CPAP.

6)    Please add more information in statistical analysis (normality etc)

7)    Please add the limitations section before conclusions

Author Response

Thank you very much for your valuable suggestion. Below are our responses.

Comments

1)      Why didn’t collected data for physical activity?

This is an important issue, indeed. Thank you for this valuable observation. Unfortunately, in our cardio-respiratory unit is not in the routine practice to collect data about physical activity. But we will mention in the text your suggestion (Section2. Materials and Methods- Study subjects)

2)    Why didn’t assessment the left ventricular internal dimension, left ventricular posterior wall, interventricular septum thickness.

These measurements are performed in current practice and provides valuable information about the HF etiology but in this study were enrolled patients with heart failure, regardless of the underlying cardiac disease. We studied patients regarding LVEF only, as main cardiac marker. We included this observation in Section2. Materials and Methods- cardiac evaluation.

3)    It would be beneficial to have data about duration of the minimum SpO2 during PSG study.

Excellent observation. We performed cardio-respiratory polygraphy for all patients and the devices permitted the collection of data only regarding longest duration SpO2 < 88% (sec) without significant differences between the groups of patients, as shown in Table 2.

We included this observation in Section2. Materials and Methods- sleep study.

4)    Which percent of participants had diabetes mellitus?

Diabetes mellitus was observed in 52.9% of patients with HFrEF, 72.7% of patients with HFmrEF and 40.2% of patients with HFpEF.  We included this information in the text, Section 3.5 Comorbidities.

5)    Please add the sleep duration and the percent of patients which used CPAP.

Because we did not performed polysomnography, sleep duration can not be clearly established.

Approximately one third of the patients included in this study used CPAP due to the non-reimbursement of the cost of this therapy. Therefore, data related to the use of CPAP have not been included in this study.

We included this observation in Section2. Materials and Methods- sleep study.

6)    Please add more information in statistical analysis (normality etc)

Statistical data

We included the comments of our statistician in the text:

Data are presented as proportions, medians and interquartile range (IQR) for variables with a skewed distribution. The differences in the characteristics of the subjects were evaluated after were divided into three groups, depending on the EF (EF<40%, EF=40-49%, EF≥50%).  We used the chi-squared test (two degrees of freedom) for comparison of categorical data between groups of patients. Continuous data were tested for normality using the Kolmogorov-Smirnov test. Data with non-normal distribution were compared using the Kruskal-Wallis test.  The P values for all hypothesis tests were two-sided, and the p value was set to the statistical significance threshold <0.005. All data analyses were performed with Stata 15.1 (Statacorp, Texas, USA).

 7)    Please add the limitations section before conclusions

 Study limitation

Thank you for your observation. We included the following comments, as suggested:

This study has several limitations. The studied population is relatively small, and even smaller for the subjects with HFrEF. There are no data about sleep, since we did not perform full-night assisted polysomnography. The results need to be confirmed by larger studies.

As recommended, we improved the introduction (section 1). We adjusted the design (Section 2), the results (Section 3), discussions (Section 4) and study limitations (Section 5). We included all the changes in resubmitted manuscript and we use "Track Changes" function in Microsoft Word, for a better visualization of the changes we performed.

We asked for the technical support from the services provided by the journal, for improved visualization of the results and better editing. We included all the changes suggested by the editor in the final manuscript.

Reviewer 2 Report

Heart failure (HF) is a highly prevalent disease with many comorbidities, and sleep-disordered breathing (SDB) is one of the most common comorbidities frequently seen in elders. This observational study investigates the prevalence of several key comorbidities in older patients with both obstructive sleep apnea syndrome (OSAS) and HF, especially in those with mid-range ejection fraction (EF). Authors conclude that patients with OSAS and HF with mid-range EF have increased risk of developing chronic kidney disease, diabetes mellitus, tricuspid and aortic insufficiency. While the conclusions are interesting, I have several concerns regarding methodology and interpretation in the study.

1.     It is not clear how statistical analysis was performed by the authors. The authors stated that they use the Chi-square test for comparison of data between groups of patients. However, a Chi-square test is designed to analyze categorical data but not parametric or continuous data. It is only meant to test the probability of independence of a distribution of data. It cannot tell whether the categories are meaningful and will not tell any details about the relationship between the data. Further, the degrees of freedom should be reported in the Chi-square test.

2.     Please state when blood samples were collected.

3.     Studies show that central sleep apnea (CSA) is highly prevalent in patients with HF. Authors need to explain why the incidence of CSA is so low in HF patients as indicated in Table 2.

Author Response

Thank you very much for your valuable suggestion. Below are our responses.

Comments and Suggestions for Authors

Heart failure (HF) is a highly prevalent disease with many comorbidities, and sleep-disordered breathing (SDB) is one of the most common comorbidities frequently seen in elders. This observational study investigates the prevalence of several key comorbidities in older patients with both obstructive sleep apnea syndrome (OSAS) and HF, especially in those with mid-range ejection fraction (EF). Authors conclude that patients with OSAS and HF with mid-range EF have increased risk of developing chronic kidney disease, diabetes mellitus, tricuspid and aortic insufficiency. While the conclusions are interesting, I have several concerns regarding methodology and interpretation in the study.

1.       It is not clear how statistical analysis was performed by the authors. The authors stated that they use the Chi-square test for comparison of data between groups of patients. However, a Chi-square test is designed to analyze categorical data but not parametric or continuous data. It is only meant to test the probability of independence of a distribution of data. It cannot tell whether the categories are meaningful and will not tell any details about the relationship between the data. Further, the degrees of freedom should be reported in the Chi-square test.

Thank you for your valuable suggestion. We included the comments of our statistician in the text:

Data are presented as proportions, medians and interquartile range (IQR) for variables with a skewed distribution. The differences in the characteristics of the subjects were evaluated after were divided into three groups, depending on the EF (EF<40%, EF=40-49%, EF≥50%).  We used the chi-squared test ((two degrees of freedom) for comparison of categorical data between groups of patients. Continuous data were tested for normality using the Kolmogorov-Smirnov test. Data with non-normal distribution were compared using the Kruskal-Wallis test.  The P values for all tests were two-sided, and the p value was set to the statistical significance threshold <0.005. All data analyses were performed with Stata 15.1 (Statacorp, Texas, USA).

2.       Please state when blood samples were collected.

This is an important aspect. Thank you for your observation. We included in the text, in section 2. Materials and methods- laboratory tests:

Blood samples were collected, early in the morning, after fasting, within 1-2 days of informed consent if signing took place in a different time of day.

3.       Studies shows that central sleep apnea (CSA) is highly prevalent in patients with HF. Authors need to explain why the incidence of CSA is so low in HF patients as indicated in Table 2.

Excellent observation, indeed. You are absolutely right. We are aware of the high prevalence of CSA in HF patients but intentionally we included only OSA patients and excluded all predominantly CSA patients in the final analysis. We included this observation in section 2. Materials and methods- Exclusion criteria.

As recommended, we improved the introduction (section 1). We adjusted the design (Section 2), the results (Section 3), discussions (Section 4) and study limitations (Section 5). We included all the changes in resubmitted manuscript and we use "Track Changes" function in Microsoft Word, for a better visualization of the changes we performed.

We ask for the technical support from the services provided by the journal, for improved visualization of the results and better editing. We included all the changes suggested by the editor in the final manuscript.

Round 2

Reviewer 1 Report

Thanks for responses